# Effect of behavioral interventions on schistosomiasis-related knowledge, attitudes, and practices of schoolchildren in Pemba, Tanzania: A 4-year repeated cross-sectional study

**Naomi Chi Ndum**[1,2], **Lydia Trippler**[1,2,3], **Ulfat Amour Mohammed**[4], **Mohammed Nassor Ali**[4], **Jan Hattendorf**[1,2], **Shaali Makame Ame**[5], **Fatma Kabole**[5], **Jürg Utzinger**[1,2], **Said Mohammed Ali**[4], **Stefanie Knopp**[1,2]*

1 Swiss Tropical and Public Health Institute, Allschwil, Switzerland, 2 University of Basel, Basel, Switzerland, 3 University of Glasgow, Glasgow, United Kingdom, 4 Public Health Laboratory–Ivo de Carneri, Chake Chake, Pemba, United Republic of Tanzania, 5 Ministry of Health, Zanzibar Town, Unguja, United Republic of Tanzania

* s.knopp@swisstph.ch

## Abstract

### Background

Schistosomiasis is a parasitic disease that mostly affects school-age children in sub-Saharan Africa. Preventive chemotherapy is the mainstay of control. Other interventions, including behavior change communication (BCC), are recommended to reduce transmission and ultimately achieve elimination. We determined the effect of BCC interventions on schistosomiasis-related knowledge, attitudes, and practices (KAP) among children who were exposed to different BCC intervention frequencies and durations within the 4-year SchistoBreak project in Pemba, Tanzania.

### Methodology

Annual surveys were carried out from 2020 to 2024 in 18 primary schools. Schistosomiasis-related KAP were assessed in randomly selected children attending grades 3–5. BCC interventions were implemented for one period (4 schools), two periods without gap (3 schools), two periods with a 1-year gap (1 school), or never (10 schools). Linear and logistic mixed-models with random effect were applied to assess associations between BCC exposure categories as predictors and knowledge or attitude scores, or unsafe washing practices as the outcome variable in 2024.

### Principal findings

A total of 4196 children participated in the surveys. Knowledge and attitude improved with continuous or repeated exposure to BCC interventions. In 2024, the knowledge scores were significantly higher in children who received BCC intervention once

**Data availability statement:** All relevant data are within the manuscript and its Supporting Information files.

**Funding:** Funding for the study has been obtained from the Swiss National Science Foundation (SNSF; Bern, Switzerland) via a PRIMA grant (PR00P3_179753) to SK. The salaries of SK and LT were funded through this PRIMA grant. Moreover, NCN is supported by a personal stipend from the Swiss Government Excellence Scholarships (ESKAS) program and the Swiss Tropical and Public Health Institute (Swiss TPH). The funders had no role in study design, data collection and analysis, decision to publish, or preparation of the manuscript.

**Competing interests:** The authors have declared that no competing interests exist.

(difference: 1.2, 95% confidence interval (CI): 0.6-1.9) or twice (difference: 2.1, 95% CI: 1.4-2.7), as compared to children who never received BCC interventions. Children who were exposed to BCC interventions twice (difference: 1.2, 95% CI: 0.8-1.6) had significantly higher attitude scores in 2024. Washing practices improved regardless of whether or not children were exposed to BCC. The washing platforms installed in intervention areas were known by up to half of the children exposed to BCC interventions, but their use was considerably lower, varying between 7.5% and 43.1%.

## Conclusions/significance

We demonstrated that BCC interventions were able to improve schistosomiasis-related knowledge and attitudes in schoolchildren. Repeated BCC interventions might be required for sustainable long-term impact.

## Trial registration

ISRCTN, ISRCTN91431493. Registered 11 February. 2020, https://www.isrctn.com/ISRCTN91431493

### Author summary

Urogenital schistosomiasis is a parasitic disease that affects mainly children in resource-constrained communities that have no access to improved water supply. On the Zanzibar islands in Tanzania, urogenital schistosomiasis is targeted for elimination. To support this goal, besides other interventions, behavior change communication (BCC) measures were implemented in the SchistoBreak project from 2020 to 2024. Eighteen primary schools in the study area received interactive BCC for varying periods. Overall, 4196 children participated in annual surveys to assess their schistosomiasis-related knowledge, attitude, and practices (KAP). Knowledge and attitude improved with continuous or repeated exposure to BCC interventions. In 2024, the knowledge and attitude scores were significantly higher in children who had received BCC interventions twice, as compared with children who never received BCC interventions. Washing practices improved, regardless of whether or not children were exposed to BCC. The safe washing platforms installed in intervention areas were known by up to half of the children exposed to BCC interventions, but their use was much lower (7.5-43.1%). Improved KAP is important to support schistosomiasis elimination through behavior change. For a lasting effect, repeated BCC interventions are necessary, and they should be coupled with access to improved water infrastructure.

## Introduction

Schistosomiasis is a neglected tropical disease caused by parasitic blood flukes of the genus *Schistosoma*. The disease mainly occurs in tropical and subtropical

countries, with the highest disease burden concentrated in sub-Saharan Africa [1,2]. Individuals of all age are susceptible to *Schistosoma* infection, but school-age children are at highest risk, explained by their behavior of frequently using infested freshwater bodies [3,4]. Indeed, schistosomiasis transmission occurs at freshwater bodies where children play, bathe, or fetch water for domestic chores [5].

Preventive chemotherapy with praziquantel is the global strategy to control schistosomiasis morbidity [6]. Year after year, praziquantel is administered to millions of individuals in endemic countries, particularly to school-age children [7]. Preventive chemotherapy programs implemented over the past 20 years in sub-Saharan Africa have contributed to a substantial decrease in the prevalence and intensity of infection, and hence, morbidity due to schistosomiasis [8]. In addition to preventive chemotherapy, the World Health Organization (WHO) recommends environmental interventions, improved access to water, sanitation, and hygiene (WASH), and behavior change measures to reduce transmission in endemic settings [6]. Such an integrated intervention approach can also lead to interruption of transmission [9,10]. Moreover, environmental and behavioral factors are considered essential to prevent the rebound of *Schistosoma* infections, alongside preventive chemotherapy programs and snail control measures [6].

Behavior change communication (BCC) is employed to deliver tailored messages concerning a particular subject to a target population, with the objective of initiating positive changes in behavior [11]. For example, BCC can be used to improve people's health literacy [11]. Indeed, better disease-related knowledge can change attitudes, resulting in the adoption of sustainable healthier behaviors and practices [12]. Since *Schistosoma* infection and transmission occur at freshwater bodies, BCC intervention strategies for schistosomiasis prevention are aimed at reducing or stopping people's water contact behaviors [13]. Hence, these strategies may include BCC for promoting the use of protective equipment (e.g., wearing boots), the adoption of ground games for children (e.g., tug of war, hopscotch, and jump rope), improving WASH, and enhanced uptake of treatment by at-risk communities [14–17].

On the Zanzibar islands, periodic preventive chemotherapy campaigns with praziquantel targeting school-age children have been implemented since the early 2000s [18–20]. As a result, the prevalence of *Schistosoma haematobium* and morbidity due to urogenital schistosomiasis were substantially reduced [20]. Subsequently, in 2010, efforts to eliminate urogenital schistosomiasis as a public health problem and to interrupt transmission were intensified [21]. In addition to preventive chemotherapy provided in schools and communities across the islands, snail control and BCC measures were implemented in selected schools and communities as part of specific research projects [21–23]. In the SchistoBreak project, conducted from 2020 to 2024, *S. haematobium* hotspot areas in the North of Pemba received a combination of preventive chemotherapy, snail control, and BCC interventions, while low-prevalence areas were subjected to a surveillance-response approach [23].

The study presented here focuses on the BCC interventions implemented within the 4-year SchistoBreak project across three intervention periods. Specifically, this study aimed to determine the effect of BCC interventions on schistosomiasis-related knowledge, attitudes, and practices (KAP) of schoolchildren that were exposed to different frequencies and durations of BCC interventions. Moreover, we assessed a potential association of the BCC exposure frequency with KAP or *S. haematobium* infection, as determined in the final survey of the SchistoBreak project in 2024.

## Methods

### Ethics statement

Before the onset of the SchistoBreak project, the study protocol received a waiver from the ethics committee of Northwestern and Central Switzerland (EKNZ; project-ID: Req-2019–00951). The study protocol was granted ethical approval by the Zanzibar Health Research Institute (ZAHRI) in December 2019 (ZAHREC/03/PR/December/2019/12), which was renewed annually. The final extension was approved by ZAHRI in March 2023 (ZAHREC/04/AMEND/MARCH/2023/03). The study was prospectively registered for an International Standard Randomised Controlled Trial Number (ISRCTN; https://www.isrctn.com/ISRCTN91431493).

The local researchers informed potential study participants about the purpose and procedures of the study. Upon recruitment, children invited to participate were provided with an information sheet and informed consent form (ICF) for their parents/legal guardians to read and sign, respectively. Only children who submitted a written ICF signed by their parents/legal guardians were included in the study. Additionally, children aged 12–17 years were invited to provide written signed assent for their participation.

Kiswahili was the main language used for communication during information meetings, informed consent procedures, questionnaire surveys, and implementation of BCC intervention activities. All documents used for local application in the study were translated from English into Kiswahili with the support of the research team and validated prior to use.

Each participant who tested positive for microhematuria and/or was infected with *S. haematobium* was informed about their diagnostic result and offered 40 mg/kg praziquantel for treatment together with biscuits and juice, or porridge.

## Study area and population

The Zanzibar islands are a semi-autonomous part of the United Republic of Tanzania. The population is predominantly Muslim, and Kiswahili and English are the two official languages [24].

The SchistoBreak project was conducted on Pemba Island from 2020 to 2024 [23]. According to the national census of 2022, Pemba had a population of 543,500 inhabitants [24]. Pemba is divided into four districts; namely, Chake Chake, Micheweni, Mkoani, and Wete, which are further subdivided into smaller administrative areas, known as shehias. The SchistoBreak project area consisted of 20 shehias in Wete and Micheweni districts. Pipe-borne water and wells were available to 54.5% and 34.5% of households, respectively [28]. For agricultural, domestic, minor industrial purposes, and some recreational activities, people rely mainly on groundwater such as rivers and streams, as tap water is often interrupted [28–30]. In 2020, 26 primary schools and 239 Islamic schools (madrassas) were located in the study area [25]. Children attend madrassas in addition to primary or secondary schooling [25–27]. The current study was conducted among children attending public primary schools in the study area.

## Study design

In the SchistoBreak project, a longitudinal design was employed with four annual school-based and community-based surveys that used a cross-sectional sampling approach and three annual intervention periods [23]. Participants' parasitologic data were collected alongside schistosomiasis-related KAP data. Based on parasitologic data, shehias were classified either as hotspot (*S. haematobium* prevalence ≥3% in schoolchildren and/or ≥2% in community members) or as low-prevalence areas (*S. haematobium* prevalence <3% in schoolchildren and <2% in community members). In the periods between the annual surveys, interventions for schistosomiasis elimination were implemented (Fig 1). Hotspot shehias received a combination of at least annual preventive chemotherapy in schools and communities, snail control, and BCC interventions [23,31,32]. Low-prevalence areas were subjected to a targeted surveillance-response approach [23,31]. This manuscript mainly focuses on the BCC interventions implemented in hotspot shehias, and on the schistosomiasis-related KAP and *S. haematobium* infections in children attending grades 3–5 of all primary schools included in the study.

## BCC interventions

The BCC interventions in our study consisted of three intervention components. The first component was community-based. It included the installation of two washing platforms per hotspot shehia and the organization of several community meetings. The washing platforms were constructed at a place that the study team had identified together with the community near an improved water source (e.g., well, pump, or tap) and aimed to provide a safe alternative to washing dishes or clothes at unimproved open freshwater bodies (e.g., rivers, streams, ponds, or lakes), where *S. haematobium* might be transmitted. The community meetings aimed to inform about the transmission of *S. haematobium*, symptoms of urogenital schistosomiasis, preventive strategies, and treatment.

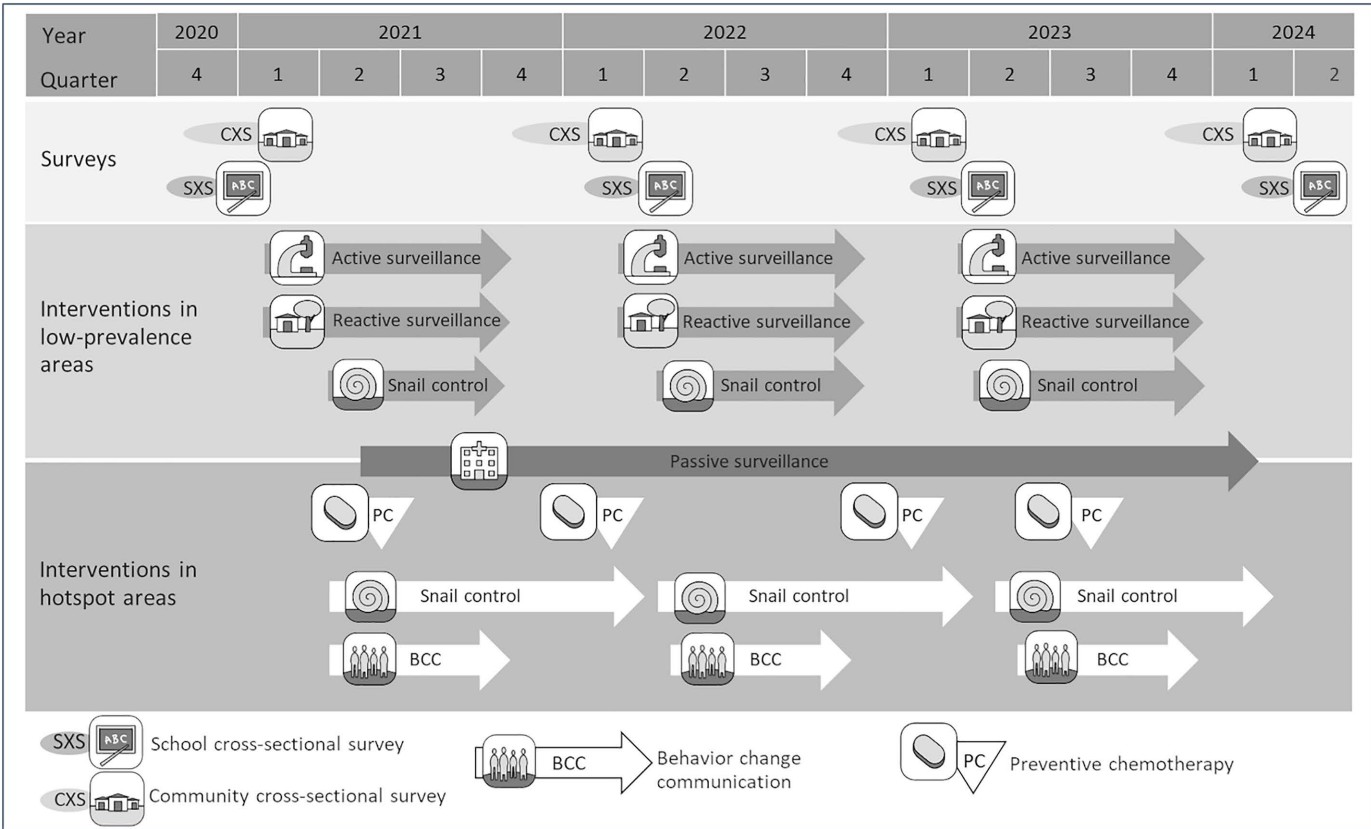

**Fig 1. Parasitologic surveys and schistosomiasis elimination interventions in low-prevalence and hotspot areas of the SchistoBreak project in Pemba, Tanzania, from 2020 to 2024.**

The second intervention component was school-based and focused on interactive classroom teaching. Teachers from public primary schools and madrassas were trained to educate children about schistosomiasis transmission and prevention using flip charts, snail boards, blood fluke pictures, life cycle drawing, and safe play methods [23,33].

The third intervention component was also school-based and targeted the whole school by health communication days called "Kichocho days" (note: kichocho is the Kiswahli term for schistosomiasis) [23,33]. At Kichocho days, children watched and listened to dramas, songs, and games about schistosomiasis that their schoolmates performed in front of the whole school. Children learned about morbidity caused by urogenital schistosomiasis, the life cycle of *S. haematobium*, and measures to prevent an infection.

## Cross-sectional school surveys

The annual school-based surveys were conducted in 18 public primary schools [25]. Children from nursery and grades 1–6 were eligible to participate in the parasitologic surveys, while children attending grades 3–5 were additionally eligible to participate in the KAP surveys. If a school had more than one class per grade, a computer-generated randomization list was employed to select either class A, B, C, or D, respectively. Subsequently, in each selected class per grade, 25 children were randomly selected for participation, using the following approach: all children in a class queued up, stratified by sex, and every third child in each line was systematically selected to be included in the study until 25 children were recruited [23]. Details of the sample size calculations are provided in the published study protocol [23]. The children were

registered electronically and assigned a unique identifier (ID) code using tablet computers (Samsung Galaxy Tab A; Samsung Electronics, Seoul, South Korea, 2019) where Open Data Kit (ODK, www.opendatakit.org) software was installed.

In the parasitologic survey, children with signed ICF (and assent) were given a plastic container tagged with a QR code entailing their ID to provide a fresh urine sample. In the KAP surveys, participants were interviewed in private with a pre-tested questionnaire (S1 Text) using ODK. Participants' age, sex, and shehia of residence were recorded and information about their schistosomiasis-related KAP were collected.

## Laboratory procedures

Urine samples were produced by the children between 10:00 and 14:00 hours in school. Samples were transferred to the Public Health Laboratory–Ivo de Carneri (PHL-IdC) in Chake Chake, Pemba, on the same day. In the laboratory, urine samples were subjected to a filtration method. For this purpose, each sample was mixed vigorously and 10 ml of urine were taken up with a syringe. A filter holder (Millipore-Merck KGaA; Darmstadt, Germany) containing a 13 mm diameter fabric filter (Sefar Ltd.; Bury, United Kingdom) was attached to the syringe and the urine was filtered. The filter was placed on a microscope slide, covered with hydrophilic cellophane soaked in glycerol, stained with Lugol's iodine and examined under a light microscope for *S. haematobium* eggs. The presence and number of *S. haematobium* eggs were recorded on paper case report forms.

## Data management

The registration and KAP questionnaire data were entered directly by the field enumerators into ODK using tablet computers and subsequently sent to a secure ODK server at the Swiss Tropical and Public Health Institute (Swiss TPH) in Allschwil, Switzerland. The laboratory data were recorded on paper case report forms by laboratory technicians. Data were double-entered into a Microsoft Excel database version 2016 (Microsoft Cooperation; Redmond Washington, United States of America) by two data entry clerks at the PHL-IdC and uploaded to a secure server at Swiss TPH. The data were checked for discrepancies using the statistical software R version 4.4.0 (www.r-project.org). Errors were traced back to the original paper data forms and corrections were applied.

## Statistical analysis

Data were analyzed using R version 4.4.0 (S1 and S2 Data). Only data from children attending grades 3–5 were included into the statistical analyses described here.

Schools participating in the study were stratified into BCC exposure categories: they were either not exposed to BCC at all, or exposed to BCC for one intervention period, for two intervention periods without a gap, or for two intervention periods with a 1-year gap.

Participants' sociodemographic data (i.e., sex, age, and school grade) were analyzed descriptively and presented as frequencies and percentages. Data from the KAP questionnaires were analyzed as follows: each of the three knowledge and two attitude multiple-choice questions were scored as incorrect (0 points), partially correct (0.5 points), or correct (1 point) with a cumulative highest possible score of 13 for knowledge and 9 for attitude, respectively, per annual survey. Subsequently, Jenks natural break was employed to classify the respective cumulative scores across all four surveys into distinct categories by minimizing variance within each category and maximizing variance between categories. Knowledge about schistosomiasis was classified into four categories: no knowledge (0 points), poor (1.0-1.5 points), moderate (2.0-2.5 points), or good (3.0-8.0 points) knowledge. Attitude was classified into three categories: poor (0-0.5 points), moderate (1.0-1.5 points), or good (2.0-5.5 points).

Practices for schistosomiasis prevention were analyzed descriptively from three questions about the main source of water children used for washing their bodies, dishes or clothes. Furthermore, data from the first and final surveys

conducted at the start and end of the SchistoBreak project in 2020 and 2024, were used to generate box plots, which illustrate the distribution of knowledge and attitude scores of children who were exposed to BCC for different intervention periods.

To assess a potential association of the number of intervention period categories (never, once, twice without gap, or twice with a 1-year gap) with knowledge or attitudes of children, respectively, as categorical variables, a linear mixed-model with random effects was employed, adjusting for age and sex, and including the school as cluster variable. Of note, the categories for three schools that were exposed to BCC interventions twice without gap and for the one school that was exposed to BCC interventions twice with a 1-year gap were combined to gain enough clusters for the analyses. The results were presented as estimated difference and 95% confidence intervals (CIs) using forest plots. Statistical significance was considered when the 95% CI of the difference did not include 0.

To assess a potential association of the number of intervention period categories (never, once, twice without gap, and twice with a 1-year gap) with washing practices or *S. haematobium* infection, respectively, as binary variables, a logistic mixed-model with random effects was used, adjusting for age and sex, and including school as cluster variable. Washing practices were stratified into safe (if a child used either tap or well water for all washing practices) or unsafe (if a child used river/pond water for any washing practice). Participants were considered as *S. haematobium*-positive if at least one *S. haematobium* egg was detected in 10 ml of their urine. Of note, the categories for three schools that were exposed to BCC interventions twice without gap and for the one school that was exposed to BCC interventions twice with a 1-year gap were combined to gain enough clusters for the analyses. The results were presented as odds ratio (OR) including 95% CIs using forest plots. Statistical significance was considered when the 95% CI did not include 1 for OR.

To ensure consistent handling of missing data (coded in R as "NA" [not available]) across the variables used in the models, conditional logic was applied. Hence, by default in R, the statistical analysis excluded all observations in variables containing missing data.

## Results

### Study participants

Over the whole SchistoBreak project, a total of 4626 children attending grades 3–5 in 18 schools were registered to participate in the study. Among them, 2852 children visited 10 schools that never received BCC interventions, 864 were from four schools that received BCC interventions in one period, 632 were from three schools that received BCC interventions in two consecutive periods, and 278 were from the remaining school that received BCC in two non-consecutive periods, separated by a 1-year gap (Fig 2). Among the registered children, 79 did not participate in the KAP questionnaires.

From the 4547 schoolchildren who participated in the KAP questionnaire surveys, only those that attended a school without BCC interventions, or who completed surveys immediately before or after a BCC intervention were included in the analysis. Hence, this resulted in KAP data from 2804 children with no BCC exposure, 632 children exposed to BCC once, 487 children exposed to BCC twice in consecutive periods, and 273 children exposed to BCC twice with a 1-year gap. Among those groups, 10, 8, 2, and 1 children, respectively, lacked urine filtration data.

In the 10 schools that never received the BCC interventions, 716 children participated in the KAP questionnaire survey in 2020, 660 in 2022, 725 in 2023, and 703 in 2024. In the four schools that received the BCC interventions for one period, 183 children were interviewed before and 218 children after the intervention. Additionally, 231 children were interviewed after the BCC interventions had stopped for 1 year. In the three schools that received the BCC interventions for two periods without a gap, 139 children were interviewed before the first, 161 children after the first and before the second intervention period, and 187 children after the second intervention period. In the school that received the interventions twice with a 1-year gap, 66 children participated before the first BCC intervention, 65 after the first intervention, 71 after the 1-year gap, and 71 after the second intervention period.

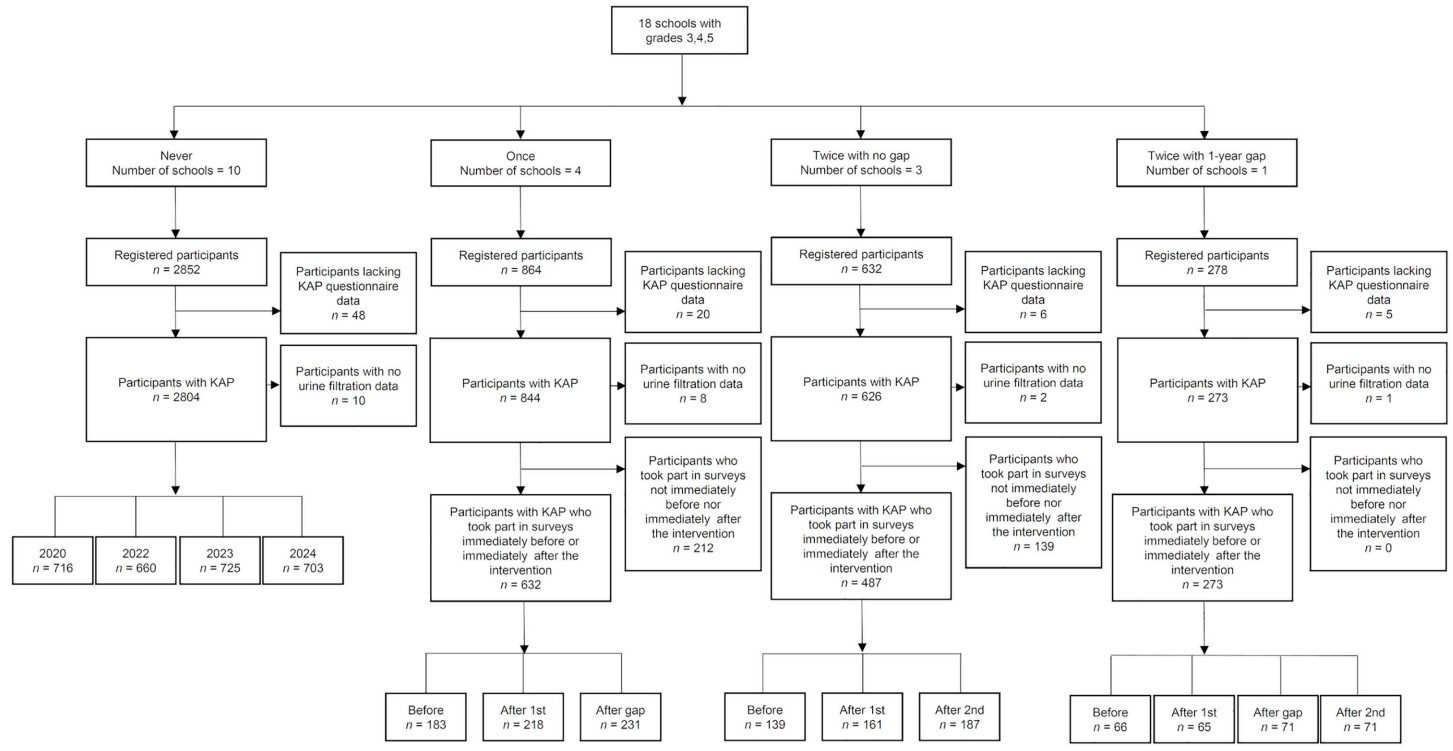

**Fig 2. Exposure categories of schools to behavior change communication (BCC) interventions and participation in surveys from 2020–2024.** Schools participation in BCC intervention periods and participation of children in questionnaire interviews about their schistosomiasis-related knowledge, attitude, and practices (KAP), and urine examinations for determining *Schistosoma haematobium* infection in Pemba, Tanzania, from 2020-2024.

Of the 4196 schoolchildren who were included in the statistical analyses, 2148 (51.2%) were females (Table 1). The interquartile age range was 10–12 years, with a median age of 11 years. Demographic details for each intervention exposure category group are summarized in Table 1.

## Knowledge of schistosomiasis prevention

In schools that never received BCC interventions, only a few children had good knowledge about schistosomiasis and around one-third had no knowledge in all four cross-sectional surveys (Fig 3A and S1 Table).

In schools that received BCC interventions once, about one of eight children (13.1%) demonstrated good knowledge prior to the BCC interventions, while 20.2% had no knowledge (Fig 3B). After the interventions, the percentage of children with good knowledge increased (56.4%) and only few children had no knowledge (5.0%). However, one year after the interventions ended, the percentage of children with good knowledge decreased (39.4%), while the percentage of children with no knowledge increased (13.4%).

In schools that received the BCC interventions for two consecutive periods, 18.7% children demonstrated good knowledge, while 12.9% had no knowledge before the first BCC intervention period (Fig 3C). After the first intervention period, the proportion with good knowledge increased to 47.8%, while the proportion with no knowledge declined to 3.7%. After the second intervention period, good knowledge rose further (65.2%), while the percentage of children with no knowledge remained low (6.4%).

In the school where BCC interventions were implemented twice with a 1-year gap, slightly less than one-fifth of the children (18.2%) demonstrated good knowledge, while the same percentage had no knowledge (18.2%) before the first

**Table 1. Characteristics of children participating in questionnaire interviews about their schistosomiasis-related knowledge, attitude, and practices (KAP) in 18 schools in Pemba, Tanzania, that were either not exposed to behavior change communication (BCC), or exposed to BCC for a different frequency of intervention periods, from 2020-2024.** LIQR: lower interquartile range; UIQR: upper interquartile range.

| Variable | Response | Total (n=4196) | | Never received BCC interventions (n=2804) | | Received BCC interventions once (n=632) | | Received BCC interventions twice with no gap (n=487) | | Received BCC interventions twice with a 1-year gap (n=273) | |
|---|---|---|---|---|---|---|---|---|---|---|---|
| | | N | % | N | % | N | % | N | % | N | % |
| **Sex** | Female | 2148 | 51.2 | 1447 | 51.6 | 320 | 50.6 | 243 | 49.9 | 138 | 50.5 |
| | Male | 2048 | 48.8 | 1357 | 48.4 | 312 | 49.4 | 244 | 50.1 | 135 | 49.5 |
| **Age (years)** | Median (LIQR - UIQR) | 11 (10-12) | | 11 (10-12) | | 11 (10-12) | | 11 (10-12) | | 11 (10-12) | |
| **School grade** | 3 | 1414 | 33.7 | 920 | 32.8 | 227 | 35.9 | 179 | 36.8 | 88 | 32.2 |
| | 4 | 1414 | 33.7 | 943 | 33.6 | 214 | 33.9 | 164 | 33.7 | 93 | 34.1 |
| | 5 | 1368 | 32.6 | 941 | 33.6 | 191 | 30.2 | 144 | 29.5 | 92 | 33.7 |

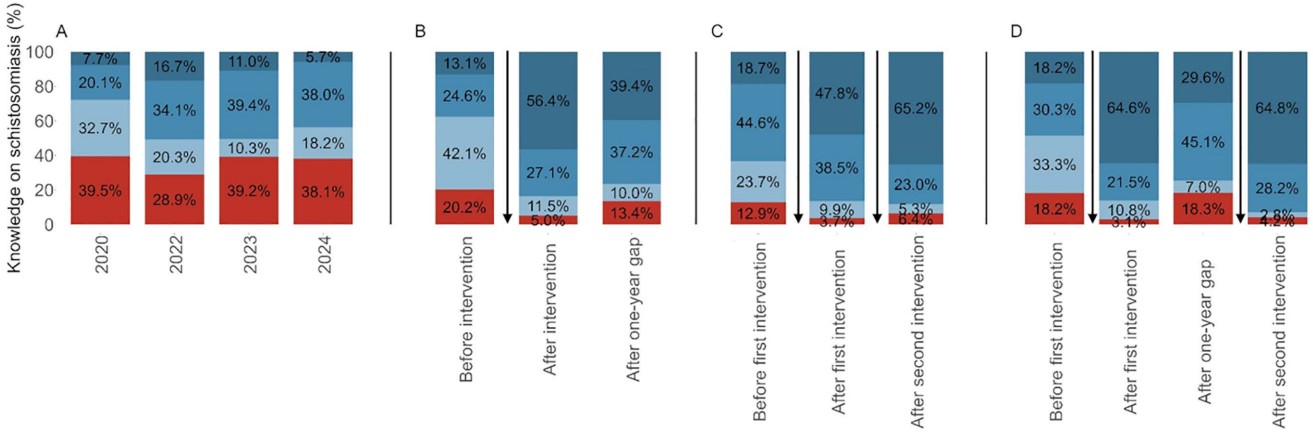

**Fig 3. Schistosomiasis-related knowledge among children from 18 schools in Pemba, Tanzania.** Schistosomiasis-related knowledge among children who were either not exposed to behavior change communication (BCC) (A; n = 10 schools), or exposed to BCC for one intervention period (B; n = 4 schools), for two intervention periods without a gap (C; n = 3 schools), or for two intervention periods with a 1-year gap (D; n = 1 school).

BCC intervention (Fig 3D). After the first intervention period, good knowledge rose sharply to about two-thirds (64.6%), while the population with no knowledge dropped to 3.1%. After a 1-year gap of BCC interventions, the trend was somewhat reverted: the percentage of children with good knowledge decreased (29.6%) and the percentage of children with no knowledge increased (18.3%). After the second period of BCC interventions was implemented, good knowledge rebounded to 64.8% and the percentage of children with no knowledge declined (4.2%).

## Attitude toward schistosomiasis prevention

Among the schools that were not part of BCC interventions, only a small proportion children exhibited a good attitude about schistosomiasis prevention and transmission, whereas more than half of the children had a poor attitude in all four cross-sectional surveys (Fig 4A and S1 Table).

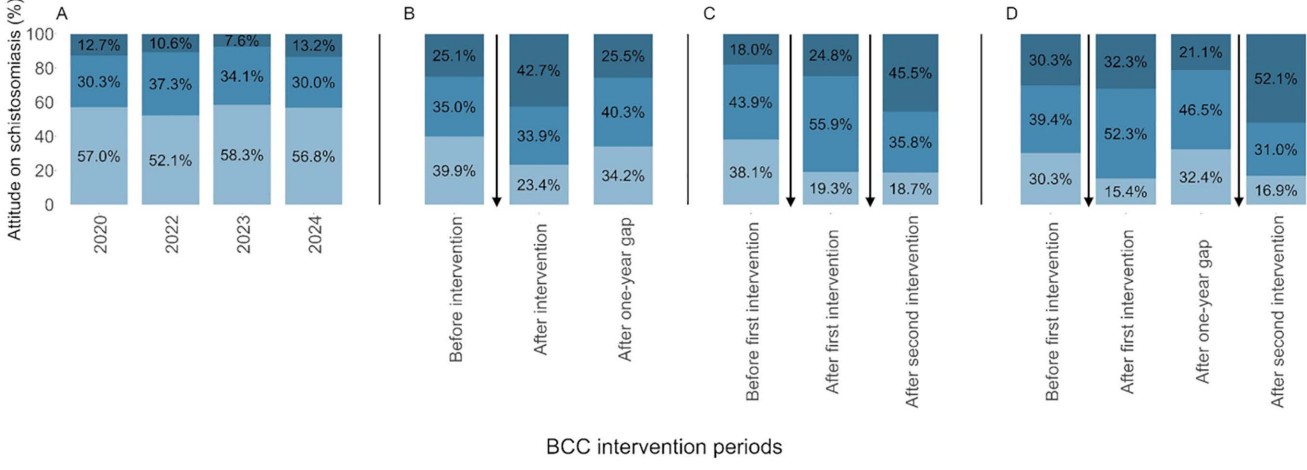

**Fig 4. Schistosomiasis-related attitude among schoolchildren from 18 schools in Pemba, Tanzania.** Schistosomiasis-related attitude among schoolchildren from 18 schools who were either not exposed to behavior change communication (BCC) (A; n = 10 schools), or exposed to BCC for one intervention period (B; n = 4 schools), for two intervention periods without a gap (C; n = 3 schools), or for two intervention periods with a 1-year gap (D; n = 1 school).

In schools that received BCC interventions once, one-quarter of the children (25.1%) showed a good attitude, while two-fifth of the children (39.9%) demonstrated a poor attitude prior to the BCC interventions (Fig 4B). Following the interventions, the proportion of children with good attitudes increased (42.7%), while the percentage of children with poor attitudes decreased (23.4%). This trend was reversed a year after the BCC intervention ended with a sharp decline in the proportion of children with a good attitude (25.5%), and an increasing number of children with poor attitude (34.2%).

In schools that received the BCC interventions twice without a gap, around one-fifth of the children (18.0%) had a good attitude, and two-fifths of the children (38.1%) had a poor attitude before the first BCC intervention period (Fig 4C). After the first intervention period, the percentage of children with a good attitude increased (24.8%), whereas poor attitude decreased (19.3%). This trend continued and after the second intervention period, almost half of the children had a good attitude (45.5%), while the percentage of children with poor attitudes decreased (18.7%).

Furthermore, in schools with two non-consecutive BCC interventions about one-third of the children (30.3%) had a good attitude and one-third of the children (30.3%) had a poor attitude prior to the first BCC intervention period (Fig 4D). After the first intervention period, the good attitude was maintained (32.3%) and the poor attitude decreased (15.4%). After a 1-year gap of BCC interventions, the percentage of children with a good attitude decreased (21.1%) and the percentage of children with poor attitude increased (32.4%). Once the second period of BCC interventions was implemented, the trend was reverted, and more than half of the children had a good attitude (52.1%), while the percentage of children with a poor attitude decreased (16.9%).

### Use of water from open freshwater bodies for domestic activities

In schools that never received the BCC interventions, the percentage of children who reported using water from unimproved open freshwater bodies for washing clothes or dishes, or were bathing, decreased progressively from the first to the third cross-sectional survey and then increased in the final survey (Fig 5A).

In schools that received the BCC interventions once, the percentage of children who reported carrying out household chores at unimproved open freshwater bodies decreased for all activities after the BCC interventions (Fig 5B).

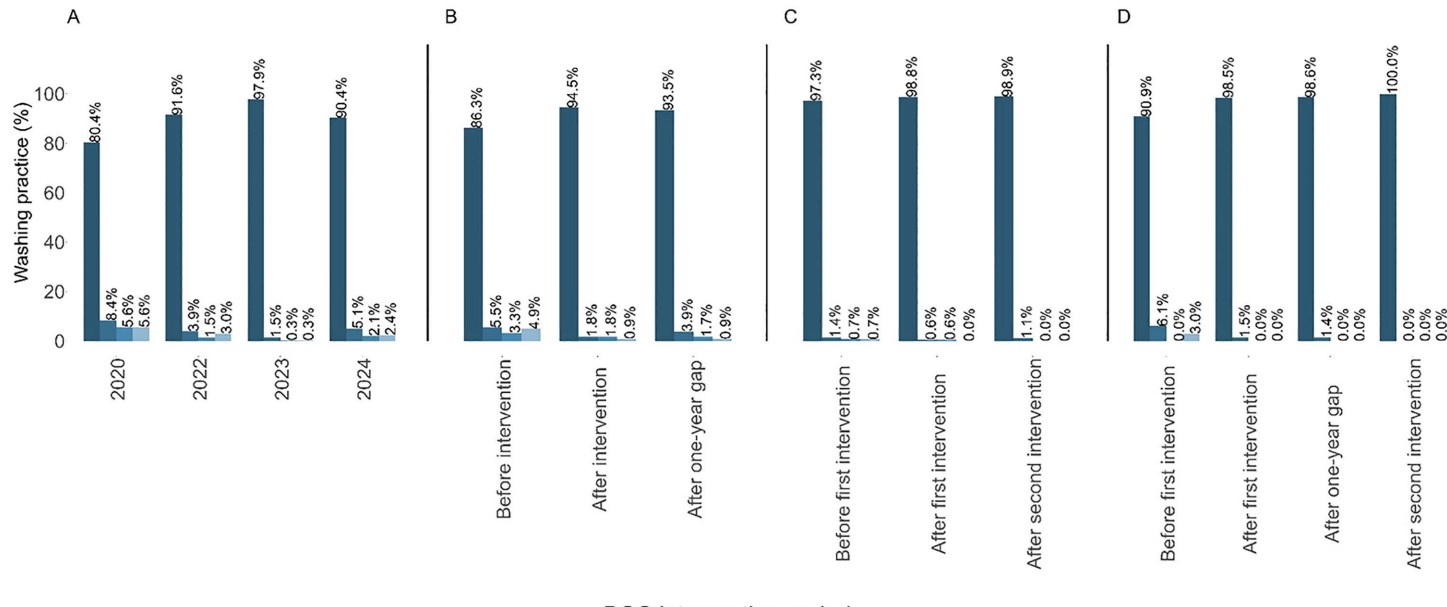

**Fig 5. Water-related practices that exposed schoolchildren from 18 schools in Pemba, Tanzania, to *Schistosoma haematobium* transmission.** Schoolchildren who were either not exposed to behavior change communication (BCC) (A; n = 10 schools), or exposed to BCC for one intervention period (B; n = 4 schools), for two intervention periods without a gap (C; n = 3 schools), or for two intervention periods with a 1-year gap (D; n = 1 school).

After a 1-year gap of BCC interventions, the percentage of children who conducted their chores at ponds and rivers had increased.

In schools that received BCC interventions twice without a gap, the percentage of children who reported washing clothes or dishes and were bathing at unimproved open freshwater bodies decreased progressively (Fig 5C). This practice was maintained throughout the second intervention and beyond, with no case of water contact activities in freshwater bodies.

Children from schools with two non-consecutive BCC interventions were observed to engage in washing and bathing in freshwater bodies prior to BCC intervention (Fig 5D). However, following the intervention, these activities ceased, extending beyond the second BCC intervention period.

## Awareness and use of washing platforms

Two washing platforms were installed in each hotspot shehia, in the year when a shehia met the crieria for a hotspot for the first time. In contrast, no washing platforms were installed in those shehias that were not classified as hotspots.

In this regard, children from schools that were not part of BCC interventions showed poor awareness and low usage of washing platforms. In 2020, very few children (1.1%) were aware of the platforms, and no usage was reported. In 2022, awareness rose slightly to 1.7%, with 0.6% of children reporting usage. In 2023 and 2024, awareness and usage were negligible, with rates at or near zero (Table 2).

Also a few children (1.1%) from one-time BCC intervention schools showed knowledge of the washing platforms with no usage prior to the intervention. After the intervention period, the percentage of children who knew (24.3%) about the

**Table 2. Awareness and use of washing platforms among children from 18 schools in Pemba, Tanzania, who were either not exposed to behavior change communication (BCC), or exposed to BCC for a different frequency of intervention periods, from 2020–2024.**

| | Knowledge of washing platforms | | Use of washing platforms | |
|---|---|---|---|---|
| | n/N | % | n/N | % |
| **Never received BCC interventions** | | | | |
| 2020 | 8/716 | 1.1 | 0/726 | 0.0 |
| 2022 | 11/660 | 1.7 | 4/660 | 0.6 |
| 2023 | 0/725 | 0.0 | 0/725 | 0.0 |
| 2024 | 2/703 | 0.3 | 0/703 | 0.0 |
| **Received BCC interventions once** | | | | |
| Before intervention | 2/183 | 1.1 | 0/183 | 0.0 |
| After intervention | 53/218 | 24.3 | 27/218 | 12.4 |
| After 1-year gap | 25/231 | 10.8 | 11/231 | 4.8 |
| **Received BCC interventions twice without gap** | | | | |
| Before first intervention | 7/139 | 5.0 | 0/139 | 0.0 |
| After first intervention | 55/161 | 34.2 | 39/161 | 24.2 |
| After second intervention | 47/187 | 25.1 | 14/187 | 7.5 |
| **Received BCC interventions twice with a 1-year gap** | | | | |
| Before first intervention | 8/66 | 12.1 | 0/66 | 0.0 |
| After first intervention | 38/65 | 58.5 | 28/65 | 43.1 |
| After 1-year gap | 17/71 | 23.9 | 9/71 | 12.7 |
| After second intervention | 23/71 | 32.4 | 7/71 | 9.9 |

washing platforms and who used (12.4%) them increased. After the BCC interventions ended, the percentage of children who knew (10.8%) and used (4.8%) washing platforms had decreased.

Similarly, children from schools with two consecutive BCC interventions had minimal knowledge (5.0%) of the washing platforms with no usage recorded prior to the intervention. After the first intervention period, the percentage of children who knew (34.2%) and used (24.2%) the washing platforms increased. After the second intervention period, the percentage of children who knew (25.1%) and used (7.5%) the washing platform decreased.

In schools that received the BCC interventions twice with a 1-year gap, several children knew (12.1%) about washing platforms before the BCC interventions started, but no child used them. After the first BCC intervention period, more than half of the children knew (58.5%), but less than half used (43.1%) the washing platforms. After a 1-year gap, the percentage of children who knew (23.9%) and used (12.7%) the washing platforms decreased. After the second intervention period, the percentage of children who knew (32.1%) about washing platforms increased, but the percentage of those who used (9.9%) the washing platforms decreased further.

### Schistosomiasis-related knowledge and attitude in the final year of the SchistoBreak project

Fig 6A shows that in the final survey of the SchistoBreak project conducted in 2024, schistosomiasis-related knowledge in children who were exposed to BCC interventions was considerably better than in children who were never exposed to BCC interventions. Arithmetic mean and median knowledge scores increased with repeated and extended exposure to BCC interventions.

Fig 6B indicates that the arithmetic mean and median attitude scores were higher in children who were exposed to BCC interventions compared with those who were never exposed to BCC interventions. Arithmetic mean attitude scores increased with increasing exposure frequency and time.

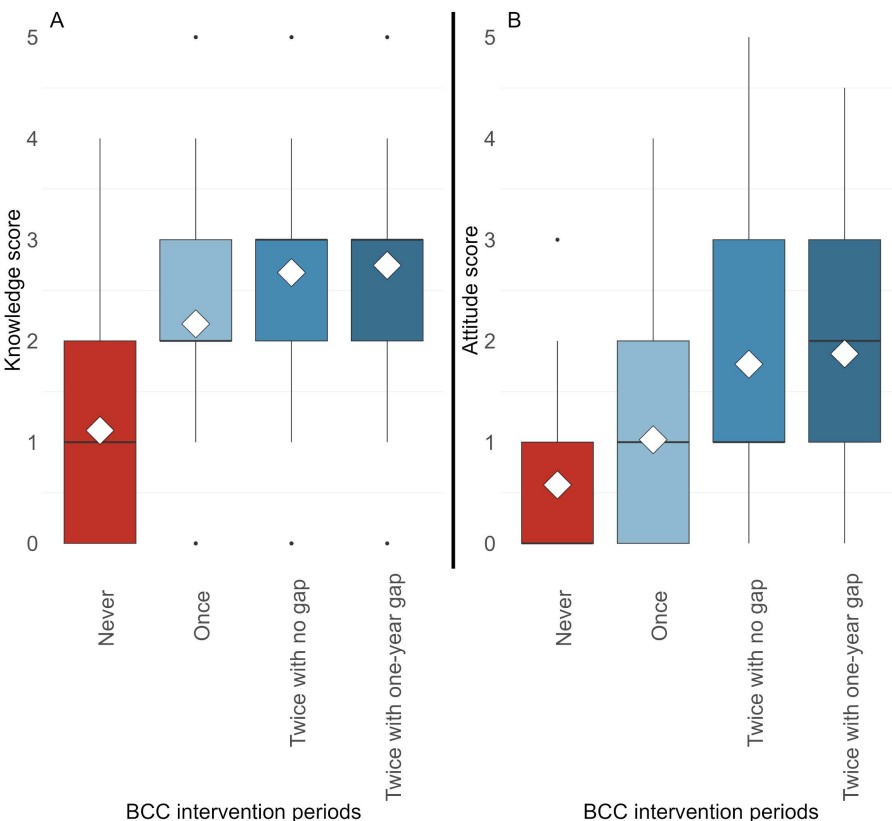

**Fig 6. Distribution of knowledge scores (A) and attitude scores (B) of children from 18 schools in Pemba, Tanzania, who were either not exposed to behavior change communication (BCC) (n=10 schools), or exposed to BCC for one intervention period (n=4 schools), for two intervention periods without a gap (n=3 schools), or for two intervention periods with a 1-year gap (n=1 school) in the final survey of the SchistoBreak project in 2024.** The box plots show minimum, maximum, median, and interquartile ranges. The white diamond represents the arithmetic mean.

### Association of BCC exposure with KAP and *S. haematobium* infection

In the final survey of the SchistoBreak project conducted in 2024, the knowledge scores about schistosomiasis were significantly higher in children who received the BCC intervention once (difference: 1.21, 95% CI: 0.58–1.85) or twice (difference: 2.08, 95% CI: 1.44–2.71), compared with children that were not part of BCC interventions. Boys had significantly higher knowledge scores (difference: 0.14, 95% CI: 0.00–0.27) than girls (Fig 7A). Increasing age had an inverse significant relationship (difference: -0.05, 95% CI: -0.10–0.00) with knowledge scores.

A similar trend was observed in attitude scores. Children who received BCC interventions once (difference: 0.47, 95% CI: 0.04–0.90) or twice (difference: 1.21, CI: 0.77–1.63) had significantly higher attitude scores compared with children who never received BCC interventions. Boys had significantly higher attitude scores (difference: 0.14, 95% CI: 0.03–0.24) than girls (Fig 7B). Increasing age had an inverse significant relationship (difference: -0.11, 95% CI: -0.15– -0.07) with attitude scores.

For practices on the use of unimproved water sources (Fig 7C) or for harboring a *S. haematobium* infection (Fig 7D), no differences in odds were observed between children who received BCC once or twice compared to children who never received BCC, and between boys and girls.

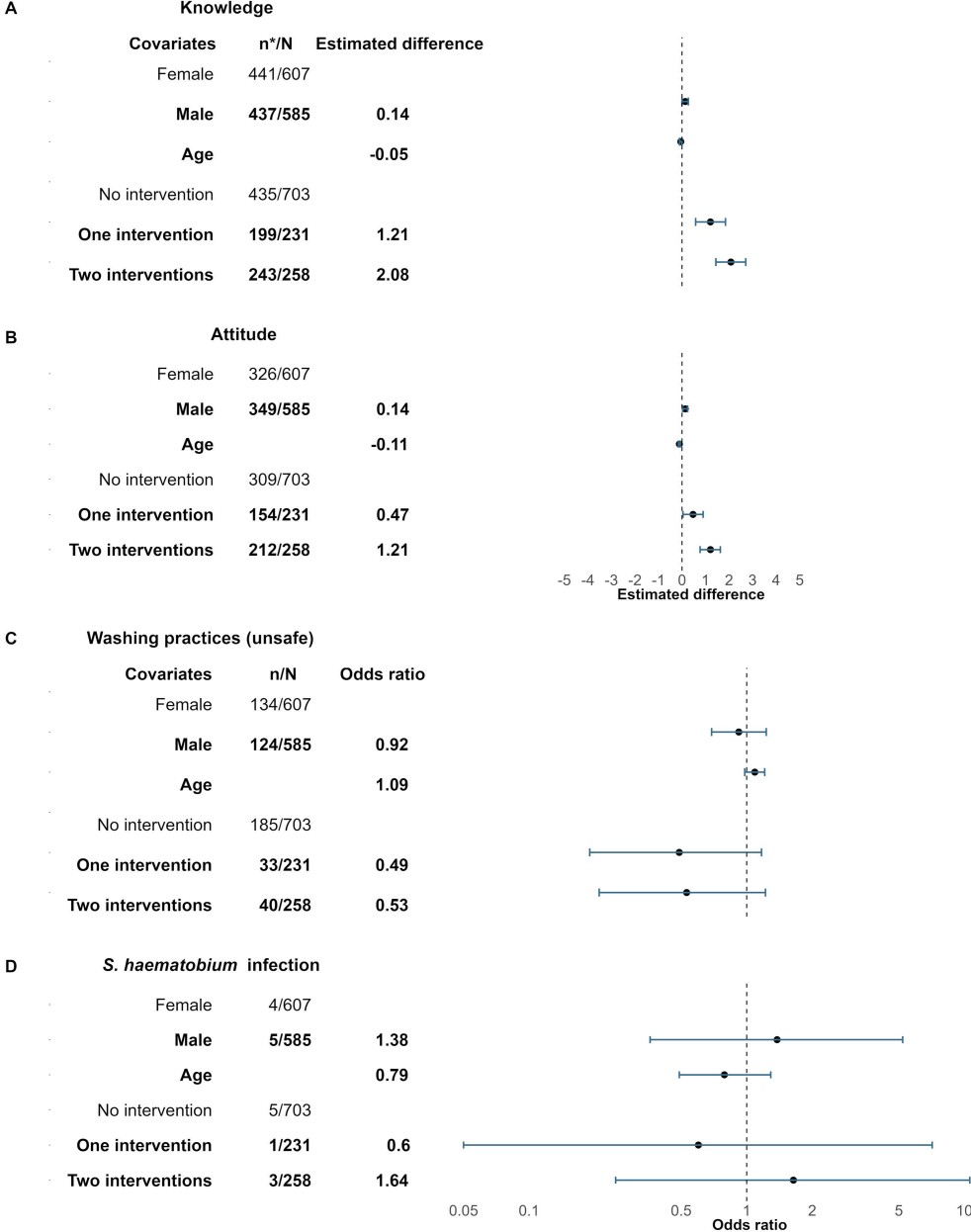

**Fig 7. Association between behavior change communication (BCC) exposure frequency and knowledge score, attitude score, practices and *Schistosoma haematobium* infection.** Association between BCC exposure frequency using a linear mixed-model for children's knowledge **(A)** and attitude **(B)**, and using a logistic mixed-model for children's washing practices **(C)** and *S. haematobium* infection **(D)**, as assessed in the final survey of the SchistoBreak project in Pemba, Tanzania, in 2024. Note that n*/N represents the proportion of children with knowledge or attitude scores >0, respectively and that n/N represents the proportion of children with unsafe water use parctices or *S. haematobium* infection, repsectively. The blue line shows the 95% confidence intervals (CIs). OR: odds ratio.

## Discussion

Schistosomiasis primarily affects school-age children in sub-Saharan Africa [3,4]. Preventive chemotherapy is the mainstay of the global schistosomiasis control strategy and has substantially reduced the prevalence and intensity of infection, and hence, morbidity [1]. Whenever resources allow, BCC measures are recommended to further reduce transmission, prevent the rebound of infections, and contribute to the ultimate goal of disease elimination [6,9,10]. We determined the effect of BCC interventions on schistosomiasis-related KAP among schoolchildren who were exposed to BCC for a varying number of intervention periods during the 4-year SchistoBreak project implemented in Pemba, an area targeted for urogenital schistosomiasis elimination.

Our study showed that knowledge increased with repeated BCC intervention exposure. Moreover, children who received BCC interventions once or twice had considerably better schistosomiasis-related knowledge than those who never received BCC interventions at the end of the study. Good knowledge increased with repeated exposure. However, once BCC interventions were halted or interrupted for a certain period, the percentage of children with good knowledge decreased.

Interestingly, children from BCC intervention schools had more schistosomiasis-related knowledge preceding the interventions than children from non-intervention schools. Since the former schools were located in hotspot areas, where schistosomiasis has been a public health problem for a long time, children were probably more aware of the disease and its transmission than their counterparts from low-prevalence areas. Many children that were not part of the BCC interventions had no knowledge about schistosomiasis and the existing levels of knowledge remained relatively stable over time. These findings illustrate that knowledge acquired through BCC interventions is improved and accumulated over time and that long-term and repeated exposure to BCC is needed for a lasting effect. A positive impact of long-term BCC was also observed in Mozambique, where community members showed improved knowledge about schistosomiasis after receiving a 1-year community dialogue intervention [35]. Moreover, studies from Brazil and a prior study in Zanzibar, where schistosomiasis-related BCC interventions were implemented for 2 and 4 years, respectively, showed that long-term exposure can have beneficial effects in creating awareness for preventive behaviors among children [13, 36]. Similar findings were reported in Ethiopia, where BCC interventions had a significant impact on improving knowledge and practices for other neglected tropical diseases, like soil-transmitted helminthiasis and onchocerciasis [12].

The BCC interventions applied in our study also had an obvious impact on the schistosomiasis-related attitude among the children. Children who received BCC interventions once or twice had significantly higher attitude scores than unexposed children in the final survey. Good and moderate attitude, which was reflected in knowledge about behaviors that reduce the risk of exposure to or transmission of *S. haematobium* also increased over time with continuous or repeated implementation of BCC interventions. The attitudes that were assessed in our study were about behaviors that can prevent *S. haematobium* infection and transmission, including not urinating, playing, washing, or swimming in the river, but instead using clean water for washing, playing elsewhere, and taking treatment. Definitively, there is room for children to learn more and understand better what behaviors are needed to prevent infection and transmission. Future BCC interventions may need to emphasize these aspects more forcefully. As discussed elsewhere and revealed by our study, learning and comprehension require repeated exposure and take time, just as behavior change does [37, 38]. Moreover, sustainable behavior change not only depends on time but also on the availability of viable alternatives to the use of open freshwater sources, such as access to improved WASH infrastructure [34, 37, 39].

A substantial decline in the use of open freshwater bodies for washing clothes or dishes and bathing was found in children visiting schools of all investigated BCC exposure categories. Additionally, a profound decrease was observed after the first intervention period and children who were exposed to BCC measures for two periods hardly reported the use of any natural open water bodies after the second intervention. However, a decrease was also detected in children from schools that were never exposed to BCC interventions, except for the final survey of our study. Hence, while children who were exposed to BCC interventions had likely adopted safer washing practices in line with what they learned, it may also be that the general water infrastructure in the North of Pemba was improved, as reported from other areas in Zanzibar [40].

In the hotspot shehias of our study area, but not in the low-prevalence areas, our study team together with the local communities, had installed two washing platforms near taps or wells as alternative options to ponds and rivers for washing clothes and dishes. Up to half of the children from hotspot schools that were exposed to BCC reported knowing about the washing platforms after the intervention period when platforms had been installed. The knowledge was partially lost when the BCC interventions were ceased or paused. The reported usage was low and related to BCC exposure in the preceding period and to the novelty of the platforms. These results point to the importance of keeping community members, including children, actively engaged in the promotion of the use of safe alternatives, and in their maintenance, to achieve long-term sustainability of WASH infrastructure in order to fight against neglected tropical diseases like schistosomiasis [14, 17].

This study has several strengths and limitations. A key strength and novel aspect is that the new BCC intervention approach was investigated in a 4-year study, allowing to assess its effect on schistosomiasis-related KAP across different periods of implementation. Evidence was generated that documented the gain or loss of KAP with extended periods of intervention or when interventions were paused or stopped, respectively. The longitudinal design over 4 years adds robustness to the study outcomes. Moreover, the random selection of participants enhanced generalizability of our findings, and the use of mixed models accounted for intra-school variability. Limitations of the study include the discrepancy in sample sizes in each intervention period. Especially the very small number of participants and that only one school was included in the category that received BCC intervention twice with a 1-year gap limits the comparative power. Weaknesses also include the already existing schistosomiasis-related knowledge and attitude that children who attended hotspot schools had prior to the onset of our BCC interventions. Of note, after adjusting for the respective mean knowledge score and attitude score per school at baseline, the interpretation of our results in the final survey did not change. The random sampling approach used in the cross-sectional annual surveys did not allow to account for the change in knowledge and attitude scores for individual study participants between the surveys in 2020 and 2024. Hence, future studies may consider larger sample sizes in all comparison groups, a longitudinal design with cohorts consisting of the same individuals and assessing the effect of BCC in a "naïve" population. Moreover, barriers to behavioral adoption such as cultural norms or infrastructural issues may be explored in more detail and using qualitative research methods. Finally, monitoring the actual use of WASH infrastructure and not just awareness will be important.

In our study, knowledge and attitude about the life cycle of *S. haematobium*, places of transmission, and where and during what activities the parasite is transmitted increased with continuous or repeated exposure to BCC interventions but was partially lost after a 1-year gap in exposure or when interventions ceased altogether. Reported washing practices were not strongly linked to BCC exposure suggesting that knowledge alone is not sufficient for behavior change. For a maximum and lasting effect on KAP, BCC interventions should be applied over a long time and consolidated by improved access to WASH infrastructure in schools and communities. Behavior change toward schistosomiasis prevention and altering transmission is more likely when individuals and the whole community are repeatedly informed about the disease transmission routes and consequences and engaged in taking up preventive behavior, and when access to adequate alternatives is provided, which can ultimately support disease elimination efforts.

## Supporting information

**S1 Text. Questionnaire.**
(PDF)

**S1 Data. Minimal Data Set and Dictionary.**
(XLSX)

**S2 Data. Analysis Codes for "R".**
(R)

**S1 Table. Table and Scoring System.**
(PDF)

**S2 Table. STROBE checklist.** The filled checklist is based on the STROBE Statement-Checklist of items that should be included in reports of observational studies, developed by the STROBE Initiative, https://www.strobe-statement.org/.
(PDF)

## Acknowledgments

The authors wish to express their gratitude to all schoolchildren who participated in this study. We sincerely thank the teachers of the study schools and the team members at the Public Health Laboratory–Ivo de Carneri in Pemba, Tanzania, for their tremendous support in implementing the study activities.

## Author contributions

**Conceptualization:** Naomi Chi Ndum, Lydia Trippler, Jan Hattendorf, Stefanie Knopp.

**Data curation:** Naomi Chi Ndum, Lydia Trippler, Stefanie Knopp.

**Formal analysis:** Naomi Chi Ndum, Jan Hattendorf.

**Funding acquisition:** Jürg Utzinger, Stefanie Knopp.

**Investigation:** Lydia Trippler, Ulfat Amour Mohammed, Said Mohammed Ali, Said Ali Mohammed, Stefanie Knopp.

**Methodology:** Naomi Chi Ndum, Lydia Trippler, Jan Hattendorf, Stefanie Knopp.

**Project administration:** Said Mohammed Ali, Stefanie Knopp.

**Resources:** Shaali Makame Ame, Fatma Kabole, Jürg Utzinger, Said Mohammed Ali, Stefanie Knopp.

**Software:** Naomi Chi Ndum, Lydia Trippler, Jan Hattendorf.

**Supervision:** Lydia Trippler, Fatma Kabole, Jürg Utzinger, Said Mohammed Ali, Stefanie Knopp.

**Visualization:** Naomi Chi Ndum, Lydia Trippler, Jan Hattendorf, Stefanie Knopp.

**Writing – original draft:** Naomi Chi Ndum, Stefanie Knopp.

**Writing – review & editing:** Naomi Chi Ndum, Lydia Trippler, Jan Hattendorf, Jürg Utzinger, Stefanie Knopp.

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
