## [Decision Letter · Decision Letter 0]

12 Jun 2025

PNTD-D-25-00558

Effect of behavioral interventions on schistosomiasis-related knowledge, attitudes, and practices of school children in Pemba, Tanzania: a 4-year study

Dear Dr. Knopp,

Thank you for submitting your manuscript to PLOS Neglected Tropical Diseases. After careful consideration, we feel that it has merit but does not fully meet PLOS Neglected Tropical Diseases's publication criteria as it currently stands. Therefore, we invite you to submit a revised version of the manuscript that addresses the points raised during the review process.

Please submit your revised manuscript within 60 days Aug 11 2025 11:59PM. If you will need more time than this to complete your revisions, please reply to this message or contact the journal office at plosntds@plos.org. Please include the following items when submitting your revised manuscript:

We look forward to receiving your revised manuscript.

Kind regards,

Jean-Philippe Chippaux, M.D., Ph.D.

Academic Editor

Justin Remais

Section Editor

Shaden Kamhawi

co-Editor-in-Chief

Paul Brindley

co-Editor-in-Chief

**Journal Requirements:**

1) Some material included in your submission may be copyrighted. According to PLOSu2019s copyright policy, authors who use figures or other material (e.g., graphics, clipart, maps) from another author or copyright holder must demonstrate or obtain permission to publish this material under the Creative Commons Attribution 4.0 International (CC BY 4.0) License used by PLOS journals. Please closely review the details of PLOSu2019s copyright requirements here: PLOS Licenses and Copyright. If you need to request permissions from a copyright holder, you may use PLOS's Copyright Content Permission form.

Potential Copyright Issues:

i) Figure 1. Please confirm whether you drew the images / clip-art within the figure panels by hand. If you did not draw the images, please provide (a) a link to the source of the images or icons and their license / terms of use; or (b) written permission from the copyright holder to publish the images or icons under our CC BY 4.0 license. Alternatively, you may replace the images with open source alternatives. See these open source resources you may use to replace images / clip-art:

ii) Figure 1 appears to have been modified from a previously published figure. Please provide written permission from the copyright holder to publish this under our CC-BY 4.0 license, or remove the figure / replace the image. Please note we do not recommend using standard request forms available on Publishers' websites, as they grant single use rather than republication under an open access license.

2) We note that your Data Availability Statement is currently as follows: "All relevant data are within the manuscript and its Supporting Information files.". Please confirm at this time whether or not your submission contains all raw data required to replicate the results of your study. Authors must share the “minimal data set” for their submission. PLOS defines the minimal data set to consist of the data required to replicate all study findings reported in the article, as well as related metadata and methods (https://journals.plos.org/plosone/s/data-availability#loc-minimal-data-set-definition).

3) Please amend your detailed Financial Disclosure statement. This is published with the article. It must therefore be completed in full sentences and contain the exact wording you wish to be published.

2) If any authors received a salary from any of your funders, please state which authors and which funders..

4) Please ensure that the funders and grant numbers match between the Financial Disclosure field and the Funding Information tab in your submission form. Note that the funders must be provided in the same order in both places as well. 

**Reviewers' Comments:**

Reviewer's Responses to Questions

**Key Review Criteria Required for Acceptance?**

**Methods**

-Are the objectives of the study clearly articulated with a clear testable hypothesis stated?

-Is the study design appropriate to address the stated objectives?

-Is the population clearly described and appropriate for the hypothesis being tested?

-Is the sample size sufficient to ensure adequate power to address the hypothesis being tested?

-Were correct statistical analysis used to support conclusions?

-Are there concerns about ethical or regulatory requirements being met?

Reviewer #1: YES

Reviewer #2: The methods used are appropriated. However, the following major concerns have emerged after review of the manuscript and will need to give more details/precisions:

1_The goal of this manuscript (Background) is unclear and will need to be addressed (Lines 104-111)

2_ Some repetition on the manuscript. So you can delete some of them. For example:

Lines 216-218: This has been summarily mentioned earlier in lines 128-133 under the ethical statement,

Lines 220-222: (and asked provided…..): this has been mentioned in lines 206-207

Lines 271, 272, 273: Please delecte "respectively"

Reviewer #3: The article addresses schistosomiasis, a parasitic disease affecting school-aged children in sub-Saharan Africa, and evaluates the effectiveness of Behavior Change Communication (BCC) interventions as a complement to preventive chemotherapy.

The article is structured as follows:

Background

Methods

Results

Conclusions

This is a conventional and appropriate format for presenting public health research.

**Results**

-Does the analysis presented match the analysis plan?

-Are the results clearly and completely presented?

-Are the figures (Tables, Images) of sufficient quality for clarity?

Reviewer #1: They are presented in an aggregated manner based on a score, which means that the detailed knowledge and behaviors that are so helpful for preparing, implementing and modifying BCC are missing. I suggest including the questionnaire as a supplementary file and adding a table showing the proportions of each type of knowledge or behavior over time.

Reviewer #2: Study participants is unclear and need to be written clearly. Please see some suggestions on the manuscript.

Idem with _Knowledge of schistosomiasis prevention and _ Attitude toward schistosomiasis prevention

Reviewer #3: Setting and population: 18 primary schools in Pemba, Tanzania.

Timeframe: 2020 to 2024.

Sample: 4,196 children in grades 3 to 5, randomly selected.

Intervention groups:

One BCC intervention period (4 schools)

Two BCC periods consecutively (3 schools)

Two BCC periods with a 1-year gap (1 school)

No intervention (10 schools)

Statistical analysis: Linear and logistic mixed models with random effects to control for clustering.

Strengths of Methodology:

Longitudinal design over four years adds robustness.

Random selection of participants enhances generalizability.

Use of mixed models accounts for intra-school variability.

Potential Limitations:

Only one school in the "two periods with 1-year gap" group limits comparative power.

No qualitative data included to explore reasons behind behavioral outcomes.

**Conclusions**

-Are the conclusions supported by the data presented?

-Are the limitations of analysis clearly described?

-Do the authors discuss how these data can be helpful to advance our understanding of the topic under study?

-Is public health relevance addressed?

Reviewer #1: YES

Reviewer #2: The conclusions are supported by the data presented but will need some ajustment

Reviewer #3: Strengths:

Demonstrates that education can positively affect knowledge and attitudes.

Reinforces the importance of repeated health education for sustained impact.

Uses reliable analytical models for complex school-level data.

Weaknesses:

Behavioral change (washing practices) was not strongly linked to BCC exposure — suggesting that knowledge alone is not sufficient for behavior change.

Lack of insight into barriers to behavior adoption, such as cultural norms or infrastructure issues.

**Editorial and Data Presentation Modifications?**

Reviewer #1: NO

Reviewer #2: (No Response)

Reviewer #3: Recommendations:

Incorporate qualitative methods in future studies to explore why children may not act on their knowledge.

Combine BCC with structural interventions to improve water, sanitation, and hygiene (WASH) infrastructure.

Tailor messages and tools to promote sustained behavioral change, not just awareness.

**Summary and General Comments**

Reviewer #1: General comment

The manuscript is related to the impact of behavior change interventions, which are an important schistosomiasis control strategy that is often neglected by control programmes. The article is well written, but some changes are required.

Specific comments

- Title: I suggest adding the type of study to the title: 'A four-year repeated cross-sectional study'.

- Keywords: Please add SAC.

- Methods: Add a specific section on qualitative data collection and provide more details on the questions asked to the study participants.

- Results: They are presented in an aggregated manner based on a score, which means that the detailed knowledge and behaviors that are so helpful for preparing, implementing and modifying BCC are missing. I suggest including the questionnaire as a supplementary file and adding a table showing the proportions of each type of knowledge or behavior over time.

Reviewer #2: This study describes and determines the effect of behavior change communication (BCC) interventions on schistosomiasis-related knowledge, attitudes, and practices (KAP) among children in different BCC intervention frequencies and durations in order to improve schistosomiasis elimination strategies. the experiences of implementing. Authors suggested some advices for BCC implementing strategies. This is certainly and interesting alternative social control method and will need to be published after revision.

Reviewer #3: Conclusions and Policy Implications

BCC interventions are effective in improving knowledge and attitudes.

Repetition enhances the impact — one-off interventions may not be enough.

Behavior change lags behind knowledge and attitudes, indicating a need for:

Supportive environments (e.g., access to clean water and functional hygiene infrastructure).

Community-level engagement.

Monitoring actual usage, not just awareness.

PLOS authors have the option to publish the peer review history of their article (what does this mean? ). If published, this will include your full peer review and any attached files.

**Do you want your identity to be public for this peer review?** For information about this choice, including consent withdrawal, please see our Privacy Policy .

Reviewer #1: **Yes: ** Amadou Garba

Reviewer #2: No

Reviewer #3: **Yes: ** Valdir Sabbaga Amato

**Figure resubmission:**
---

## [Decision Letter · Decision Letter 1]

9 Aug 2025

Dear Dr Knopp,

We are pleased to inform you that your manuscript 'Effect of behavioral interventions on schistosomiasis-related knowledge, attitudes, and practices of school children in Pemba, Tanzania: a 4-year repeated cross-sectional study' has been provisionally accepted for publication in PLOS Neglected Tropical Diseases.

Best regards,

Benn Sartorius, PhD

Section Editor

Justin Remais

Section Editor

Shaden Kamhawi

co-Editor-in-Chief

Paul Brindley

co-Editor-in-Chief

Reviewer's Responses to Questions

**Key Review Criteria Required for Acceptance?**

**Methods**

-Are the objectives of the study clearly articulated with a clear testable hypothesis stated?

-Is the study design appropriate to address the stated objectives?

-Is the population clearly described and appropriate for the hypothesis being tested?

-Is the sample size sufficient to ensure adequate power to address the hypothesis being tested?

-Were correct statistical analysis used to support conclusions?

-Are there concerns about ethical or regulatory requirements being met?

Reviewer #2: After revision, the objectives of the study are clearly articulated with the good hypothesis. The study population is appropriate and clearly described with the sufficient sample size. The statistical analysis used is correct and support the conclusions.

**Results**

-Does the analysis presented match the analysis plan?

-Are the results clearly and completely presented?

-Are the figures (Tables, Images) of sufficient quality for clarity?

Reviewer #2: The paper is well written with the results clearly and completely presented.

**Conclusions**

-Are the conclusions supported by the data presented?

-Are the limitations of analysis clearly described?

-Do the authors discuss how these data can be helpful to advance our understanding of the topic under study?

-Is public health relevance addressed?

Reviewer #2: The helpful conclusions are supported by the data. Public health relevance is addressed accordingly.

**Editorial and Data Presentation Modifications?**

Reviewer #2: The manuscipt is well revised and could be considerated as publication in PlosNTD

**Summary and General Comments**

Reviewer #2: The manuscipt is well revised and could be considerated as publication in PlosNTD

PLOS authors have the option to publish the peer review history of their article (what does this mean? ). If published, this will include your full peer review and any attached files.

**Do you want your identity to be public for this peer review?** For information about this choice, including consent withdrawal, please see our Privacy Policy .

Reviewer #2: **Yes: ** IBIKOUNLE Moudachirou

---

## [Editor Report · Acceptance letter]

Dear Dr Knopp,

We are delighted to inform you that your manuscript, "Effect of behavioral interventions on schistosomiasis-related knowledge, attitudes, and practices of schoolchildren in Pemba, Tanzania: a 4-year repeated cross-sectional study," has been formally accepted for publication in PLOS Neglected Tropical Diseases.

Best regards,

Shaden Kamhawi

co-Editor-in-Chief

Paul Brindley

co-Editor-in-Chief
